# Inferring Algorithmic Patterns with Stack-Augmented Recurrent Nets

**Armand Joulin**
Facebook AI Research
770 Broadway, New York, USA.
ajoulin@fb.com

**Tomas Mikolov**
Facebook AI Research
770 Broadway, New York, USA.
tmikolov@fb.com

## Abstract

Despite the recent achievements in machine learning, we are still very far from achieving real artificial intelligence. In this paper, we discuss the limitations of standard deep learning approaches and show that some of these limitations can be overcome by learning how to grow the complexity of a model in a structured way. Specifically, we study the simplest sequence prediction problems that are beyond the scope of what is learnable with standard recurrent networks, algorithmically generated sequences which can only be learned by models which have the capacity to count and to memorize sequences. We show that some basic algorithms can be learned from sequential data using a recurrent network associated with a trainable memory.

## 1 Introduction

Machine learning aims to find regularities in data to perform various tasks. Historically there have been two major sources of breakthroughs: scaling up the existing approaches to larger datasets, and development of novel approaches [5, 14, 22, 30]. In the recent years, a lot of progress has been made in scaling up learning algorithms, by either using alternative hardware such as GPUs [9] or by taking advantage of large clusters [28]. While improving computational efficiency of the existing methods is important to deploy the models in real world applications [4], it is crucial for the research community to continue exploring novel approaches able to tackle new problems.

Recently, deep neural networks have become very successful at various tasks, leading to a shift in the computer vision [21] and speech recognition communities [11]. This breakthrough is commonly attributed to two aspects of deep networks: their similarity to the hierarchical, recurrent structure of the neocortex and the theoretical justification that certain patterns are more efficiently represented by functions employing multiple non-linearities instead of a single one [1, 25].

This paper investigates which patterns are difficult to represent and learn with the current state of the art methods. This would hopefully give us hints about how to design new approaches which will advance machine learning research further. In the past, this approach has lead to crucial breakthrough results: the well-known XOR problem is an example of a trivial classification problem that cannot be solved using linear classifiers, but can be solved with a non-linear one. This popularized the use of non-linear hidden layers [30] and kernels methods [2]. Another well-known example is the parity problem described by Papert and Minsky [25]: it demonstrates that while a single non-linear hidden layer is sufficient to represent any function, it is not guaranteed to represent it efficiently, and in some cases can even require exponentially many more parameters (and thus, also training data) than what is sufficient for a deeper model. This lead to use of architectures that have several layers of non-linearities, currently known as deep learning models.

Following this line of work, we study basic patterns which are difficult to represent and learn for standard deep models. In particular, we study learning regularities in sequences of symbols gen-

| Sequence generator | Example |
|---|---|
| $\{a^n b^n \mid n > 0\}$ | aab**ba**aab**bba**b**a**aaaab**bbbb** |
| $\{a^n b^n c^n \mid n > 0\}$ | aaab**bbccca**bc**a**aaaab**bbbbccccc** |
| $\{a^n b^n c^n d^n \mid n > 0\}$ | aab**bccdda**aab**bbcccddda**bc**d** |
| $\{a^n b^{2n} \mid n > 0\}$ | aab**bbba**aab**bbbbba**b**b** |
| $\{a^n b^m c^{n+m} \mid n, m > 0\}$ | aabc**cca**aabbc**ccccca**bc**c** |
| $n \in [1, k],\ X \to nXn,\ X \to=$ | $(k = 2)$ 12=**21**2122=**221**211121=**12111** |

Table 1: Examples generated from the algorithms studied in this paper. In bold, the characters which can be predicted deterministically. During training, we do not have access to this information and at test time, we evaluate only on deterministically predictable characters.

erated by simple algorithms. Interestingly, we find that these regularities are difficult to learn even for some advanced deep learning methods, such as recurrent networks. We attempt to increase the learning capabilities of recurrent nets by allowing them to learn how to control an infinite structured memory. We explore two basic topologies of the structured memory: pushdown stack, and a list.

Our structured memory is defined by constraining part of the recurrent matrix in a recurrent net [24]. We use multiplicative gating mechanisms as learnable controllers over the memory [8, 19] and show that this allows our network to operate as if it was performing simple read and write operations, such as PUSH or POP for a stack.

Among recent work with similar motivation, we are aware of the Neural Turing Machine [17] and Memory Networks [33]. However, our work can be considered more as a follow up of the research done in the early nineties, when similar types of memory augmented neural networks were studied [12, 26, 27, 37].

## 2 Algorithmic Patterns

We focus on sequences generated by simple, short algorithms. The goal is to learn regularities in these sequences by building predictive models. We are mostly interested in discrete patterns related to those that occur in the real world, such as various forms of a long term memory.

More precisely, we suppose that during training we have only access to a stream of data which is obtained by concatenating sequences generated by a given algorithm. We do not have access to the boundary of any sequence nor to sequences which are not generated by the algorithm. We denote the regularities in these sequences of symbols as *Algorithmic patterns*. In this paper, we focus on algorithmic patterns which involve some form of counting and memorization. Examples of these patterns are presented in Table 1. For simplicity, we mostly focus on the unary and binary numeral systems to represent patterns. This allows us to focus on designing a model which can learn these algorithms when the input is given in its simplest form.

Some algorithm can be given as context free grammars, however we are interested in the more general case of sequential patterns that have a short description length in some general Turing-complete computational system. Of particular interest are patterns relevant to develop a better language understanding. Finally, this study is limited to patterns whose symbols can be predicted in a single computational step, leaving out algorithms such as sorting or dynamic programming.

## 3 Related work

Some of the algorithmic patterns we study in this paper are closely related to context free and context sensitive grammars which were widely studied in the past. Some works used recurrent networks with hardwired symbolic structures [10, 15, 18]. These networks are continuous implementation of symbolic systems, and can deal with recursive patterns in computational linguistics. While theses approaches are interesting to understand the link between symbolic and sub-symbolic systems such as neural networks, they are often hand designed for each specific grammar.

Wiles and Elman [34] show that simple recurrent networks are able to learn sequences of the form $a^n b^n$ and generalize on a limited range of $n$. While this is a promising result, their model does not

truly learn how to count but instead relies mostly on memorization of the patterns seen in the training data. Rodriguez et al. [29] further studied the behavior of this network. Grünwald [18] designs a hardwired second order recurrent network to tackle similar sequences. Christiansen and Chater [7] extended these results to grammars with larger vocabularies. This work shows that this type of architectures can learn complex internal representation of the symbols but it cannot generalize to longer sequences generated by the same algorithm. Beside using simple recurrent networks, other structures have been used to deal with recursive patterns, such as pushdown dynamical automata [31] or sequenctial cascaded networks [3, 27].

Hochreiter and Schmidhuber [19] introduced the Long Short Term Memory network (LSTM) architecture. While this model was orginally developed to address the vanishing and exploding gradient problems, LSTM is also able to learn simple context-free and context-sensitive grammars [16, 36]. This is possible because its hidden units can choose through a multiplicative gating mechanism to be either linear or non-linear. The linear units allow the network to potentially count (one can easily add and subtract constants) and store a finite amount of information for a long period of time. These mechanisms are also used in the Gated Recurrent Unit network [8]. In our work we investigate the use of a similar mechanism in a context where the memory is unbounded and structured. As opposed to previous work, we do not need to "erase" our memory to store a new unit. More recently, Graves et al. [17] have extended LSTM with an attention mechansim to build a model which roughly resembles a Turing machine with limited tape. Their memory controller works with a fixed size memory and it is not clear if its complexity is necessary for the the simple problems they study.

Finally, many works have also used external memory modules with a recurrent network, such as stacks [12, 13, 20, 26, 37]. Zheng et al. [37] use a discrete external stack which may be hard to learn on long sequences. Das et al. [12] learn a continuous stack which has some similarities with ours. The mechnisms used in their work is quite different from ours. Their memory cells are associated with weights to allow continuous representation of the stack, in order to train it with continuous optimization scheme. On the other hand, our solution is closer to a standard RNN with special connectivities which simulate a stack with unbounded capacity. We tackle problems which are closely related to the ones addressed in these works and try to go further by exploring more challenging problems such as binary addition.

## 4 Model

### 4.1 Simple recurrent network

We consider sequential data that comes in the form of discrete tokens, such as characters or words. The goal is to design a model able to predict the next symbol in a stream of data. Our approach is based on a standard model called recurrent neural network (RNN) and popularized by Elman [14].

RNN consists of an input layer, a hidden layer with a recurrent time-delayed connection and an output layer. The recurrent connection allows the propagation of information through time. Given a sequence of tokens, RNN takes as input the one-hot encoding $x_t$ of the current token and predicts the probability $y_t$ of next symbol. There is a hidden layer with $m$ units which stores additional information about the previous tokens seen in the sequence. More precisely, at each time $t$, the state of the hidden layer $h_t$ is updated based on its previous state $h_{t-1}$ and the encoding $x_t$ of the current token, according to the following equation:

$$h_t = \sigma \left( U x_t + R h_{t-1} \right), \tag{1}$$

where $\sigma(x) = 1/(1 + \exp(-x))$ is the sigmoid activation function applied coordinate wise, $U$ is the $d \times m$ token embedding matrix and $R$ is the $m \times m$ matrix of recurrent weights. Given the state of these hidden units, the network then outputs the probability vector $y_t$ of the next token, according to the following equation:

$$y_t = f \left( V h_t \right), \tag{2}$$

where $f$ is the softmax function [6] and $V$ is the $m \times d$ output matrix, where $d$ is the number of different tokens. This architecture is able to learn relatively complex patterns similar in nature to the ones captured by N-grams. While this has made the RNNs interesting for language modeling [23], they may not have the capacity to learn how algorithmic patterns are generated. In the next section, we show how to add an external memory to RNNs which has the theoretical capability to learn simple algorithmic patterns.

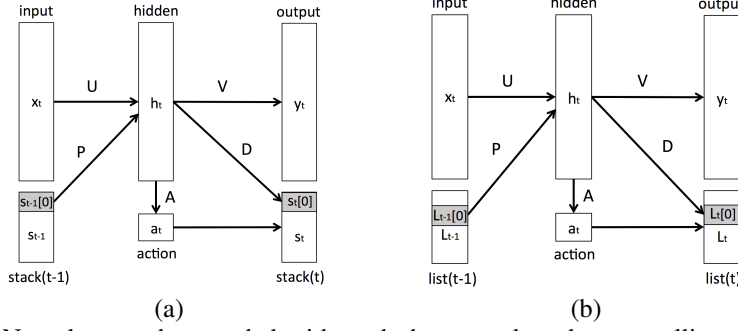

(a)                  (b)

Figure 1: (a) Neural network extended with push-down stack and a controlling mechanism that learns what action (among PUSH, POP and NO-OP) to perform. (b) The same model extended with a doubly-linked list with actions INSERT, LEFT, RIGHT and NO-OP.

## 4.2   Pushdown network

In this section, we describe a simple structured memory inspired by pushdown automaton, i.e., an automaton which employs a stack. We train our network to learn how to operate this memory with standard optimization tools.

A stack is a type of persistent memory which can be only accessed through its topmost element. Three basic operations can be performed with a stack: POP removes the top element, PUSH adds a new element on top of the stack and NO-OP does nothing. For simplicity, we first consider a simplified version where the model can only choose between a PUSH or a POP at each time step. We suppose that this decision is made by a 2-dimensional variable $a_t$ which depends on the state of the hidden variable $h_t$:

$$a_t = f\left(Ah_t\right), \tag{3}$$

where $A$ is a $2 \times m$ matrix ($m$ is the size of the hidden layer) and $f$ is a softmax function. We denote by $a_t[\text{PUSH}]$, the probability of the PUSH action, and by $a_t[\text{POP}]$ the probability of the POP action. We suppose that the stack is stored at time $t$ in a vector $s_t$ of size $p$. Note that $p$ could be increased on demand and does not have to be fixed which allows the capacity of the model to grow. The top element is stored at position 0, with value $s_t[0]$:

$$s_t[0] = a_t[\text{PUSH}]\sigma(Dh_t) + a_t[\text{POP}]s_{t-1}[1], \tag{4}$$

where $D$ is $1 \times m$ matrix. If $a_t[\text{POP}]$ is equal to 1, the top element is replaced by the value below (all values are moved by one position up in the stack structure). If $a_t[\text{PUSH}]$ is equal to 1, we move all values down in the stack and add a value on top of the stack. Similarly, for an element stored at a depth $i > 0$ in the stack, we have the following update rule:

$$s_t[i] = a_t[\text{PUSH}]s_{t-1}[i-1] + a_t[\text{POP}]s_{t-1}[i+1]. \tag{5}$$

We use the stack to carry information to the hidden layer at the next time step. When the stack is empty, $s_t$ is set to $-1$. The hidden layer $h_t$ is now updated as:

$$h_t = \sigma\left(Ux_t + Rh_{t-1} + Ps_{t-1}^k\right), \tag{6}$$

where $P$ is a $m \times k$ recurrent matrix and $s_{t-1}^k$ are the $k$ top-most element of the stack at time $t-1$. In our experiments, we set $k$ to 2. We call this model Stack RNN, and show it in Figure 1-a without the recurrent matrix $R$ for clarity.

**Stack with a no-operation.** Adding the NO-OP action allows the stack to keep the same value on top by a minor change of the stack update rule. Eq. (4) is replaced by:

$$s_t[0] = a_t[\text{PUSH}]\sigma(Dh_t) + a_t[\text{POP}]s_{t-1}[1] + a_t[\text{NO-OP}]s_{t-1}[0].$$

**Extension to multiple stacks.** Using a single stack has serious limitations, especially considering that at each time step, only one action can be performed. We increase capacity of the model by using multiple stacks in parallel. The stacks can interact through the hidden layer allowing them to process more challenging patterns.

| method | $a^n b^n$ | $a^n b^n c^n$ | $a^n b^n c^n d^n$ | $a^n b^{2n}$ | $a^n b^m c^{n+m}$ |
|---|---|---|---|---|---|
| RNN | 25% | 23.3% | 13.3% | 23.3% | 33.3% |
| LSTM | 100% | 100% | 68.3% | 75% | 100% |
| List RNN 40+5 | 100% | 33.3% | 100% | 100% | 100% |
| Stack RNN 40+10 | 100% | 100% | 100% | 100% | 43.3% |
| Stack RNN 40+10 + rounding | 100% | 100% | 100% | 100% | 100% |

Table 2: Comparison with RNN and LSTM on sequences generated by counting algorithms. The sequences seen during training are such that $n < 20$ (and $n + m < 20$), and we test on sequences up to $n = 60$. We report the percent of $n$ for which the model was able to correctly predict the sequences. Performance above 33.3% means it is able to generalize to never seen sequence lengths.

**Doubly-linked lists.** While in this paper we mostly focus on an infinite memory based on stacks, it is straightforward to extend the model to another forms of infinite memory, for example, the doubly-linked list. A list is a one dimensional memory where each node is connected to its *left* and *right* neighbors. There is a read/write head associated with the list. The head can move between nearby nodes and insert a new node at its current position. More precisely, we consider three different actions: INSERT, which inserts an element at the current position of the head, LEFT, which moves the head to the left, and RIGHT which moves it to the right. Given a list $L$ and a fixed head position HEAD, the updates are:

$$L_t[i] = \begin{cases} a_t[\text{RIGHT}]L_{t-1}[i+1] + a_t[\text{LEFT}]L_{t-1}[i-1] + a_t[\text{INSERT}]\sigma(Dh_t) & \text{if } i = \text{HEAD,} \\ a_t[\text{RIGHT}]L_{t-1}[i+1] + a_t[\text{LEFT}]L_{t-1}[i-1] + a_t[\text{INSERT}]L_{t-1}[i+1] & \text{if } i < \text{HEAD,} \\ a_t[\text{RIGHT}]L_{t-1}[i+1] + a_t[\text{LEFT}]L_{t-1}[i-1] + a_t[\text{INSERT}]L_{t-1}[i] & \text{if } i > \text{HEAD.} \end{cases}$$

Note that we can add a NO-OP operation as well. We call this model List RNN, and show it in Figure 1-b without the recurrent matrix $R$ for clarity.

**Optimization.** The models presented above are continuous and can thus be trained with stochastic gradient descent (SGD) method and back-propagation through time [30, 32, 35]. As patterns becomes more complex, more complex memory controller must be learned. In practice, we observe that these more complex controller are harder to learn with SGD. Using several random restarts seems to solve the problem in our case. We have also explored other type of search based procedures as discussed in the supplementary material.

**Rounding.** Continuous operators on stacks introduce small imprecisions leading to numerical issues on very long sequences. While simply discretizing the controllers partially solves this problem, we design a more robust rounding procedure tailored to our model. We slowly makes the controllers converge to discrete values by multiply their weights by a constant which slowly goes to infinity. We finetune the weights of our network as this multiplicative variable increase, leading to a smoother rounding of our network. Finally, we remove unused stacks by exploring models which use only a subset of the stacks. While brute-force would be exponential in the number of stacks, we can do it efficiently by building a tree of removable stacks and exploring it with deep first search.

# 5 Experiments and results

First, we consider various sequences generated by simple algorithms, where the goal is to learn their generation rule [3, 12, 29]. We hope to understand the scope of algorithmic patterns each model can capture. We also evaluate the models on a standard language modeling dataset, Penn Treebank.

**Implementation details.** Stack and List RNNs are trained with SGD and backpropagation through time with 50 steps [32], a hard clipping of 15 to prevent gradient explosions [23], and an initial learning rate of 0.1. The learning rate is divided by 2 each time the entropy on the validation set is not decreasing. The depth $k$ defined in Eq. (6) is set to 2. The free parameters are the number of hidden units, stacks and the use of NO-OP. The baselines are RNNs with 40, 100 and 500 units, and LSTMs with 1 and 2 layers with 50, 100 and 200 units. The hyper-parameters of the baselines are selected on the validation sets.

## 5.1 Learning simple algorithmic patterns

Given an algorithm with short description length, we generate sequences and concatenate them into longer sequences. This is an unsupervised task, since the boundaries of each generated sequences

| current | next | prediction | proba(next) | action | | stack1[top] | stack2[top] |
|---|---|---|---|---|---|---|---|
| b | a | a | 0.99 | POP | POP | -1 | 0.53 |
| a | a | a | 0.99 | PUSH | POP | 0.01 | 0.97 |
| a | a | a | 0.95 | PUSH | PUSH | 0.18 | 0.99 |
| a | a | a | 0.93 | PUSH | PUSH | 0.32 | 0.98 |
| a | a | a | 0.91 | PUSH | PUSH | 0.40 | 0.97 |
| a | a | a | 0.90 | PUSH | PUSH | 0.46 | 0.97 |
| a | b | a | 0.10 | PUSH | PUSH | 0.52 | 0.97 |
| b | **b** | **b** | 0.99 | PUSH | PUSH | 0.57 | 0.97 |
| b | **b** | **b** | 1.00 | POP | PUSH | 0.52 | 0.56 |
| b | **b** | **b** | 1.00 | POP | PUSH | 0.46 | 0.01 |
| b | **b** | **b** | 1.00 | POP | PUSH | 0.40 | 0.00 |
| b | **b** | **b** | 1.00 | POP | PUSH | 0.32 | 0.00 |
| b | **b** | **b** | 1.00 | POP | PUSH | 0.18 | 0.00 |
| b | **b** | **b** | 0.99 | POP | PUSH | 0.01 | 0.00 |
| b | **b** | **b** | 0.99 | POP | POP | -1 | 0.00 |
| b | **b** | **b** | 0.99 | POP | POP | -1 | 0.00 |
| b | **b** | **b** | 0.99 | POP | POP | -1 | 0.00 |
| b | **b** | **b** | 0.99 | POP | POP | -1 | 0.01 |
| b | **a** | **a** | 0.99 | POP | POP | -1 | 0.56 |

Table 3: Example of the Stack RNN with 20 hidden units and 2 stacks on a sequence $a^n b^{2n}$ with $n = 6$. $-1$ means that the stack is empty. The depth $k$ is set to $1$ for clarity. We see that the first stack pushes an element every time it sees $a$ and pop when it sees $b$. The second stack pushes when it sees $a$. When it sees $b$, it pushes if the first stack is not empty and pop otherwise. This shows how the two stacks interact to correctly predict the deterministic part of the sequence (shown in bold).

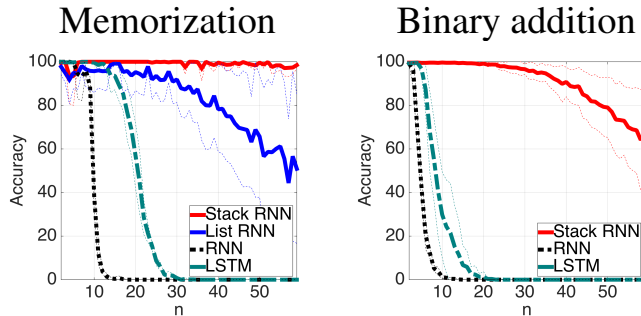

Figure 2: Comparison of RNN, LSTM, List RNN and Stack RNN on memorization and the performance of Stack RNN on binary addition. The accuracy is in the proportion of correctly predicted sequences generated with a given $n$. We use $100$ hidden units and $10$ stacks.

are not known. We study patterns related to counting and memorization as shown in Table 1. To evaluate if a model has the capacity to understand the generation rule used to produce the sequences, it is tested on sequences it has not seen during training. Our experimental setting is the following: the training and validation set are composed of sequences generated with $n$ up to $N < 20$ while the test set is composed of sequences generated with $n$ up to $60$. During training, we incrementally increase the parameter $n$ every few epochs until it reaches some $N$. At test time, we measure the performance by counting the number of correctly predicted sequences. A sequence is considered as correctly predicted if we correctly predict its deterministic part, shown in bold in Table 1. On these toy examples, the recurrent matrix $R$ defined in Eq. (1) is set to 0 to isolate the mechanisms that Stack and list can capture.

**Counting.** Results on patterns generated by "counting" algorithms are shown in Table 2. We report the percentage of sequence lengths for which a method is able to correctly predict sequences of that length. List RNN and Stack RNN have 40 hidden units and either 5 lists or 10 stacks. For these tasks, the NO-OP operation is not used. Table 2 shows that RNNs are unable to generalize to longer sequences, and they only correctly predict sequences seen during training. LSTM is able to generalize to longer sequences which shows that it is able to count since the hidden units in an LSTM can be linear [16]. With a finer hyper-parameter search, the LSTM should be able to achieve $100\%$

on all of these tasks. Despite the absence of linear units, these models are also able to generalize. For $a^n b^m c^{n+m}$, rounding is required to obtain the best performance.

Table 3 show an example of actions done by a Stack RNN with two stacks on a sequence of the form $a^n b^{2n}$. For clarity, we show a sequence generated with $n$ equal to 6, and we use discretization. Stack RNN pushes an element on both stacks when it sees $a$. The first stack pops elements when the input is $b$ and the second stack starts popping only when the first one is empty. Note that the second stack pushes a special value to keep track of the sequence length, i.e. 0.56.

**Memorization.** Figure 2 shows results on memorization for a dictionary with two elements. Stack RNN has 100 units and 10 stacks, and List RNN has 10 lists. We use random restarts and we repeat this process multiple times. Stack RNN and List RNN are able to learn memorization, while RNN and LSTM do not seem to generalize. In practice, List RNN is more unstable than Stack RNN and overfits on the training set more frequently. This instability may be explained by the higher number of actions the controler can choose from (4 versus 3). For this reason, we focus on Stack RNN in the rest of the experiments.

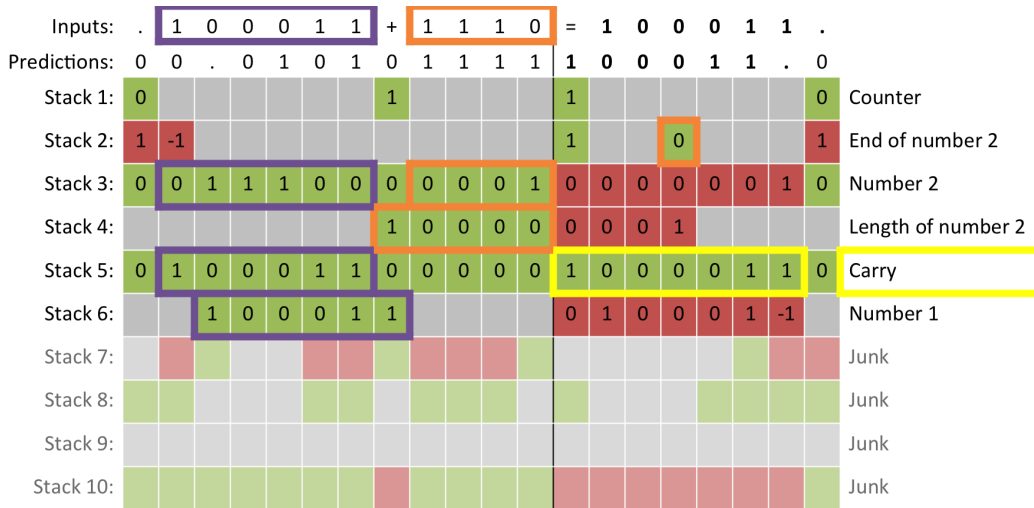

Figure 3: An example of a learned Stack RNN that performs binary addition. The last column is our interpretation of the functionality learned by the different stacks. The color code is: green means PUSH, red means POP and grey means actions equivalent to NO-OP. We show the current (discretized) value on the top of the each stack at each given time. The sequence is read from left to right, one character at a time. In bold is the part of the sequence which has to be predicted. Note that the result is written in reverse.

**Binary addition.** Given a sequence representing a binary addition, e.g., "101+1=", the goal is to predict the result, e.g., "110." where "." represents the end of the sequence. As opposed to the previous tasks, this task is supervised, i.e., the location of the deterministic tokens is provided. The result of the addition is asked in the reverse order, e.g., "011." in the previous example. As previously, we train on short sequences and test on longer ones. The length of the two input numbers is chosen such that the sum of their lengths is equal to $n$ (less than 20 during training and up to 60 at test time). Their most significant digit is always set to 1. Stack RNN has 100 hidden units with 10 stacks. The right panel of Figure 2 shows the results averaged over multiple runs (with random restarts). While Stack RNNs are generalizing to longer numbers, it overfits for some runs on the validation set, leading to a larger error bar than in the previous experiments.

Figure 3 shows an example of a model which generalizes to long sequences of binary addition. This example illustrates the moderately complex behavior that the Stack RNN learns to solve this task: the first stack keeps track of where we are in the sequence, i.e., either reading the first number, reading the second number or writing the result. Stack 6 keeps in memory the first number. Interestingly, the first number is first captured by the stacks 3 and 5 and then copied to stack 6. The second number is stored on stack 3, while its length is captured on stack 4 (by pushing a one and then a set of zeros). When producing the result, the values stored on these three stacks are popped. Finally stack 5 takes

care of the carry: it switches between two states (0 or 1) which explicitly say if there is a carry over or not. While this use of stacks is not optimal in the sense of minimal description length, it is able to generalize to sequences never seen before.

**5.2   Language modeling.**

| Model | Ngram | Ngram + Cache | RNN | LSTM | SRCN [24] | Stack RNN |
|---|---|---|---|---|---|---|
| Validation perplexity | - | - | 137 | **120** | **120** | 124 |
| Test perplexity | 141 | 125 | 129 | **115** | **115** | 118 |

Table 4: Comparison of RNN, LSTM, SRCN [24] and Stack RNN on Penn Treebank Corpus. We use the recurrent matrix $R$ in Stack RNN as well as 100 hidden units and 60 stacks.

We compare Stack RNN with RNN, LSTM and SRCN [24] on the standard language modeling dataset Penn Treebank Corpus. SRCN is a standard RNN with additional self-connected linear units which capture long term dependencies similar to bag of words. The models have only one hidden layer with 100 hidden units. Table 4 shows that Stack RNN performs better than RNN with a comparable number of parameters, but not as well as LSTM and SRCN. Empirically, we observe that Stack RNN learns to store exponentially decaying bag of words similar in nature to the memory of SRCN.

# 6   Discussion and future work

**Continuous versus discrete model and search.**   Certain simple algorithmic patterns can be efficiently learned using a continuous optimization approach (stochastic gradient descent) applied to a continuous model representation (in our case RNN). Note that Stack RNN works better than prior work based on RNN from the nineties [12, 34, 37]. It seems also simpler than many other approaches designed for these tasks [3, 17, 31]. However, it is not clear if a continuous representation is completely appropriate for learning algorithmic patterns. It may be more natural to attempt to solve these problems with a discrete model. This motivates us to try to combine continuous and discrete optimization. It is possible that the future of learning of algorithmic patterns will involve such combination of discrete and continuous optimization.

**Long-term memory.**   While in theory using multiple stacks for representing memory is as powerful as a Turing complete computational system, intricate interactions between stacks need to be learned to capture more complex algorithmic patterns. Stack RNN also requires the input and output sequences to be in the right format (e.g., memorization is in reversed order). It would be interesting to consider in the future other forms of memory which may be more flexible, as well as additional mechanisms which allow to perform multiple steps with the memory, such as loop or random access. Finally, complex algorithmic patterns can be more easily learned by composing simpler algorithms. Designing a model which possesses a mechanism to compose algorithms automatically and training it on incrementally harder tasks is a very important research direction.

# 7   Conclusion

We have shown that certain difficult pattern recognition problems can be solved by augmenting a recurrent network with structured, growing (potentially unlimited) memory. We studied very simple memory structures such as a stack and a list, but, the same approach can be used to learn how to operate more complex ones (for example a multi-dimensional tape). While currently the topology of the long term memory is fixed, we think that it should be learned from the data as well.

**Acknowledgment.**   We would like to thank Arthur Szlam, Keith Adams, Jason Weston, Yann LeCun and the rest of the Facebook AI Research team for their useful comments.

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
