[Supplementary Material · supp.pdf]

# Supplementary material: Inferring Algorithmic Patterns with Memory-Augmented Recurrent Nets

**Armand Joulin**
Facebook AI Research
770 Broadway, New York, USA.
ajoulin@fb.com

**Tomas Mikolov**
Facebook AI Research
770 Broadway, New York, USA.
ajoulin@fb.com

## 1 Search based optimization

In this section, we discuss an alternative optimization scheme for memory augmented neural networks. SGD can be seen as a greedy form of search over the space of parameters. Learning a controller of an external memory is a highly non-linear problem: single difference in memory operation can cause the algorithm to not work as intended. Despite encouraging results, we empirically observe that SGD can often get stuck in local minima.

For example, in the case of sequences of the form $a^n b^n$, the "identity" solution (i.e. a model which predicts its input) has a low entropy and is a local minima for models such as feedforward neural network. Even though the "identity" solution has an entropy which is suboptimal, models trained with SGD seems to often converge to this solution (probably due to the simplicity of the solution).

Various search based strategies can be considered to reduce this problem: the simplest one may be to include random restarts of the training mechanism while keeping in memory the best solution found so far. Another option is to include the search more directly into the learning process. As a model trained with SGD can only follow one path during its optimization (it has only one set of weights), it is possible to explore different regions of the weight space by training many different models in parallel. The exploration of the weights space can be done with a variant of beam search. The weights from the hidden layer to the output can easily be learned with gradient descent. For the action outputs we do not have the targets, as the models are trained without any supervision that would hint what should have been the correct action in a given situation.

To obtain the target for the action outputs, we use a strategy similar to reinforcement learning [1]: we sample the targets using the probability distribution over actions computed by the current model. This stochastic choice quickly makes the models different, which is what we aim for - exploring efficiently the space of parameters is important for any efficient search strategy. We continue sampling the targets for several time steps (30 in our experiments; using significantly more steps would make the models less diverse, following the law of big numbers). After that, we train all models using just targets for the output predictions for another 2000 steps. Then, we run evaluation of all models on novel data, again using 2000 steps, to see which model performs the best. All models that have below average performance are replaced in the next training epoch by the best performing model. The last trick is to keep one copy of the best model unchanged, to prevent degradation of the performance. Essentially, our optimization can be seen as beam search over actions, combined with SGD for updating the weights. We show Table 1 an example of model trained with a search+SGD scheme. Note that another advantage of this scheme is that the controller is discrete.

## 2 Results

**Supervised memorization.** We consider a supervised setting where the sequences are seperated by a delimiter ".". Figure 1, we show how our method performs when we increase the size of the dictionary in the supervised setting.

| current | next | pred | proba(next) | action | | stack1[0] | stack1[1] | stack2[0] | stack2[1] |
|---|---|---|---|---|---|---|---|---|---|
| b | a | a | 0.99 | push1 | push0 | 0 | 0 | 1 | 0 |
| a | a | a | 0.75 | push1 | push1 | 1 | 0 | 0 | 1 |
| a | a | a | 0.66 | push1 | push1 | 1 | 1 | 1 | 0 |
| a | b | b | 0.67 | push1 | push1 | 1 | 1 | 1 | 1 |
| b | b | b | 1.00 | push0 | push1 | 1 | 1 | 1 | 1 |
| b | b | b | 0.99 | push0 | pop | 0 | 1 | 1 | 1 |
| b | b | b | 0.99 | push1 | pop | 0 | 0 | 1 | 1 |
| b | b | b | 0.99 | push0 | push0 | 1 | 0 | 1 | 1 |
| b | b | b | 0.99 | push0 | pop | 0 | 1 | 0 | 1 |
| b | b | b | 0.99 | push1 | pop | 0 | 0 | 1 | 1 |
| b | b | b | 0.99 | push0 | push0 | 1 | 0 | 1 | 0 |
| b | b | b | 0.99 | push0 | pop | 0 | 1 | 0 | 1 |
| b | a | a | 0.99 | push1 | push0 | 0 | 0 | 1 | 0 |

Table 1: Example of Stack RNN with two stacks trained with SGD+search on sequences of the form $a^n b^{3n}$. The stacks stores value 0 or 1. We show the 2 first levels of the stacks. When the input is $a$, both stacks push 1, leading to $n$ 1 in both stacks. Then when the inputs become $b$, the first stack acts as a counter: it pushes the sequence 001 to count to 3. This gives the information to the other stack when we are at a multiplier of $3n$. The second stack counts to $n$, by popping a 1 at every cycle (given by the 1st stack) It also pushes 0 and pops it to simulate two no-op operations. At the end of the sequence, the second stack has $\{1, 0\}$ which means that we are at a multiplier of $n$, and the first stack has $\{0, 0\}$ which codes for a multiplier of 3.

Figure 1: Supervised memorization as the total proportion of correctly predicted sequences as a function of the dictionary size $k$. This shows that Stack RNN can memorize sequences with larger dictionary than what we studied in the main paper.

**Multiplication.** Finally we consider the problem of multiplication. We consider sequences of the form $a^n b^m c^{nm}$ for multiplication. Similarly to unsupervised memorization, we consider 10 different models and keep the 5 with the lowest entropy on the validation set. We also use the NO-OP action. Figure 2 shows their average performances on the test set. On multiplication, we see some limitation of our model. It seems that it is not able to generalize much on this task. Our model is marginally

better than RNNs but is still overfitting on the training set. The fact that our stacks are continuous is a possible reason why it overfits.

$$a^n b^m c^{nm}$$

Figure 2: On the right panel, comparison between our model and RNN on an multiplication, i.e. $\{a^n b^m c^{nm} \mid n, m > 1\}$. The accuracy is in the proportion of correctly predicted sequences generated with a fixed $n + m$. Despite having the potential to learn multiplication, the Stack RNN did prove difficult to be trained for this task.

**Comparison between SGD and search + SGD.** A simple way to break our model is to consider sequences where there is an unbalanced number of $a$s and $b$s in the sequence, e.g., $a^n b^{6n}$. This is a standard problem in supervised classification with unbalanced number of training example per class. When the statistics of the training data are known, a natural solution is to reweigh each class (e.g., tf-idf for text). However in our case, we aim at learning sequences online with potential change in their statistics, we thus cannot use such solution. On the other hand, the search algorithm described above seems to avoid this problem by exploring a larger set of possible combination of discrete actions. While our study is still preliminary, using a search algorithm on top of SGD allows us to solve sequences such as $a^n b^{4n}$ or $a^n b^{6n}$, where training our model with SGD fails.

## 2.1 Simulated world question answering

In this section, we consider the question answering tasks proposed in [2]. They propose a simulated world containing 4 characters, 3 objects and 5 rooms. At each time step, a character perform an action - moving around, picking up an object or dropping it. The actions are written into a text using simple automated grammar and every so often, a question is asked about the location of a person or an object. These tasks have been designed to test the capacity of a model to hold discrete information about this synthetic world for long period of time. There are two types of questions: either the question concerns the current location of an entity or it concerns a location previously visited. We show an example on the right panel of Table 2.

These tasks require the models to be able to store information for potentially long period of time. It also requires to store information about multiple objects and persons. Typically, this means that for our model, it will requires a large amount of stacks. These tasks require that we keep information for a long period of time, we thus use the NO-OP operation.

In Table 2, we compare our model to RNN, LSTM and the memory network (memNN, [2]). memNN is a supervised method which used annotation about the support sentences to answer the questions.

| | people | obj. | before | before+obj. | Example from "before": |
|---|---|---|---|---|---|
| Unsupervised methods | | | | | Alice went to the office, |
| LSTM (from [2]) | 100% | 70.0% | 64.8% | 49.1% | and then Bob moved to |
| RNN 100 | 99.8% | 68.3% | 59.4% | 37.3% | the garden, later Carol went |
| Ours 100-40 | 100.0% | 74.7% | 65.8% | 51.6% | to the kitchen, then Alice |
| Supervised methods | | | | | moved to the bedroom. |
| MemNN k=1 [2] | 97.8% | 30.5% | 31.0% | 24.0% | where was Alice before |
| MemNN k=2, time [2] | 100.0% | 100.0% | 100.0% | 100.0% | the bedroom? **Office**. |

Table 2: Comparison with RNN and memNN on question answering tasks of [2]. On the right, an example of question answering.

On the other hand, LSTM, RNN and our model are unsupervised. For LSTM, we use the numbers published in [2]. We see that our results are similar to the one obtained by LSTM or RNN. This suggest that our model is basically using its stacks in a similar way as the recurrent matrix of an RNN. Like LSTM, our model fails to capture long discrete patterns even in a relatively low noise setting. We think that there are two reasons for this result: first the number of combinations of entities and places in this database pushes the limit of our model representation power. Second, we think that our model struggles to store complex information required to solve these tasks. This suggests that more complex structures than stacks may be required for this type of tasks.