[Reviews · NeurIPS 2015]

Submitted by Assigned_Reviewer_1

This paper proposes an augmentation to standard recurrent nets where the net can make use of an expandable memory data structure such as a stack or list.

The idea is that the model can backpropagate through the data structure's operations to learn how to manipulate this expandable memory.

The authors show that their model can outperform standard approaches on some character sequence repetition counting tasks and that it can perform competitively on language modeling tasks.

Note that the Memory Networks work of Weston et al. and the Neural Turing Machine work of Graves et al. are strict generalizations of this paper.

I don't think that the NTM paper has been formally published, but it would seem reasonable to ask the authors to at least discuss the Memory Networks paper.

I don't think it suffices to say that this work "follows on work from the nineties" and thus you don't need to consider those papers.

I however do really like that this paper places itself well in terms of past research.

It does seem important to acknowledge that the notion of augmenting (recurrent) neural nets with datastructures is actually a rather old one.

The rest of the related work section is really appreciated.

I think something that would really help this paper would be to have more interesting experiments and motivation.

The counting task seems very specifically tailored to this particular architecture (i.e. it is trivial to count repetitions with a stack).

How would this be useful for real-world tasks?

Nitpicky things: Please don't say that your paper focuses on *infinite* memory based stacks: e.g. line 228.

This is simply completely untrue and sounds a little ingenuous.

It would be better to say that your model has expandable memory, which can grow and shrink in contrast to traditional nets that have a fixed number of hidden units.

Summary: The idea of augmenting recurrent nets with data structures and backpropagating through their operations is a really neat and super exciting direction of research.

The authors pick a nice niche in this by focusing on stacks, lists and counting operations in particular.

However, the work is simply not novel in light of e.g. the Memory Networks work of Weston et al. which is a strict generalization of this approach.

I think it's reasonable to ask that the authors at least discuss and compare to that work.

Submitted by Assigned_Reviewer_2

The authors motivate the study of problems that are generated from algorithms, as the next step in learning algorithms that goes beyond just learning of statistical regularities. The authors propose a recurrent model that uses augmented memory that is independent from the computational model itself. Specifically, the authors use RNNs (and LSTM-RNNs) to control a memory store through soft PUSH/POP actions (for stacks) (or soft INSERT/LEFT/RIGHT for doubly linked lists). The current contents of memory stores themselves are used to compute the hidden states of the RNNs, along with data and past hidden states. At each step the RNN first computes the hidden state using the memory stores and the inputs and the past hidden state. From this hidden state the next step action is generated, which is then used, along with the hidden state to update the memory store at the next time step.

Learning is done with SGD and a strategy that amounts to beam searching on actions. Results presented on toy problems show that indeed the method can perform better than RNNs and LSTMs. Results on the Penn Tree bank task on the other hand are worse than LSTMs.

My comments on the paper follow in somewhat of a random order: Some of the results are curious and unintuitive. For example the LIST RNN gets 100% on the anbncndn task but worse on the easier task of anbncn. Also anbmc(n+m) does worse then anbncndn task on Stack RNN 40+10, even though the latter are generally longer and possibly more complicated. Is this probably just a relic of the difficulty of training this models and the stochasticity in the optimization procedure described in the supplementary materials ?

The

paper does try to control overfitting on Penn Tree bank task carefully, as can be seen from the results, which are much worse than the best results on this task by Vinyals et al. As such it does disservice to the exploration of the method itself, since its not clear if its actually performing worse than LSTMs or is just overfitting more.

I wish that the optimization procedure was described in the paper itself, not shuffled off to the appendix since it is a crucial part of the paper.

It is not clear to me if the extra memory that is provided actually serves as anything other than extra computation, since memory is read /write and can be used only for the current problem. When a new instance of a problem is encountered, the memory is reset. This would be different from a model that could separate algorithms from data needed to solve a problem.

I was looking to see if the stacks had a size to them.. I would expect in practical implementation where the authors blend the stack contents from one time to the next, there would a certain size of the stack that's stored ? Is it perhaps tied to the number of time steps ?

Small note: Is the equation for linked lists updates correct ? I would have expected that insert would add an element at i, and for i < HEAD all the elements would be unchanged and copied from the last time step, whereas for i > HEAD things would be copied from i-1 elements of the last time step.. Perhaps insert is defined slightly different here, or I'm mistaken..
Summary: The paper introduces a method to learn sequential models augmented with memory. Memory can be in the form of lists or stacks and can be use to keep intermediate computations. The authors show that these models can learn algorithmic patterns better than LSTMs and RNNs; however the results on a real language task shows no gains.

Submitted by Assigned_Reviewer_3

This paper looks at extensions to RNNs in which a stack reminiscent of pushdown automata is added. This basic setup is then further expanded to include: 1) the ability to have NO-OPs (keeping the same value on the stack), 2) multiple stacks interacting through a hidden layer, and 3) the case where doubly linked lists are used.

The models explored are continuous and trainable using stochastic gradient descent (SGD); however, due to problems with local minima a search strategy based on random restarts is used for the primary experimental work.

Experiments are presented for a memorization task and a binary addition task. Experiments are also presented on the Penn Treebank.

This work is nicely presented. The paper does a reasonable job of pointing to related prior work and highlighting how this work is different. The key model variants are clear and well explained. I particularly liked Figure 3 and the explanation of the behaviours emerging within the learned stacks of the RNN trained to perform binary addition.

The paper makes the claim that in contrast to most prior work that has explored stack augmented RNNs, this work aims at exploring their potential to be used as general purpose mechanisms as opposed to focusing on their application to a specific problem. This statement is seemingly fairly true, but it is a little misleading in that Mozer and Das, (1993) did look at a fairly wide variety of settings (all cast as grammars though), whereas Zeng et al., (1994) looked at the issue through the lens of learning (general purpose) grammars. It seems the exploration of the problem of binary addition and the Penn Treebank experiments in particular do indeed nicely and clearly differentiate the work here from prior art.

I think the paper would benefit from the authors just being a little clearer about which specific aspects of their paper are particularly novel compared to prior work. I am referring in particular to the use of multiple stacks, NO-OPs and doubly linked lists and the experiments on binary addition. Has any of the cited prior work explored those particular extensions? This is a minor point, as I think being clearer about the novelty here might help increase the paper's impact. In general I think the quality of the work here is well above the acceptance threshold for NIPS and I would vote to have an oral presentation for this work.

A few other questions:

* Could you be more precise about how the discretization is performed and how sensitive the model is to the choice? It doesn't appear that this choice was a part of the formal validation set based strategy used to select parameters, but I think the reader would probably want to know how aggressive that needs to be / what threshold was used and how tricky that might or might not be.

Other minor point(s): Some minor language fixes are needed below for otherwise very interesting observation: "Empirically, we observe that [the] Stack RNN learns to store [an] exponentially decaying bag of words similar in nature to the memory of SRCN." Could you talk more about the manifestation of that in the stack structure? (Possibly in the extended version of this work)
Summary: This paper provides a good combination of quality, clarity and originality. As such, I think this work has the potential to be considered significant by the wider community and should be accepted.

Submitted by Assigned_Reviewer_4

Endowing memory to recurrent neural networks is clearly one of the most important topics of deep learning and crucial to do real reasoning. The proposed stack-augmented recurrent nets outperform simple rnn and lstm on a series of synthetic problems (learning simple algorithmic patterns). The complexity of problems is clearly defined and the behavior of resulting stack rnn could be well understood and easily analyzed. However, the conclusions merely depending on those synthetic data set may take a risk. The importance of the problems to real sequence modeling task could be uncertain and the failures of other models could be greatly improved by more and dense hyper-parameter searching. Like in (Le et al., 2015), by a very simple trick a rnn works very well on a toy task (a adding problem) which seems to need to model long term dependencies. Quoc V. Le, Navdeep Jaitly, Geoffrey E. Hinton. A SimpleWay to Initialize Recurrent Networks of Rectified Linear Units.

Minor comments In all of experiments (including language modeling), you use relative small rnn. Is the training of stack rnn very time-consuming?

How and why do you set the depth k to 2? Do you have any intuition for it? Stack rnn (a small one) does not work well on Penn tree bank. We know with dropout LSTM could get a very low perplexity. Why not use a much bigger stack rnn? Is the training is too slow? Do you observe serious overfitting?

Summary: The authors extend the capabilities of recurrent nets by coupling them to trainable and structured memory (a stack or a list), which has been show empirically modeling sequences of symbols generated by some simple context-free grammar. The proposed recurrent nets are well motivated, clearly and properly evaluated on the synthetic data set.

Author Feedback
Author rebuttal: We thank the reviewers for their thoughtful comments and address below their concerns.

R1 and R6.
Comparison with Neural Turing Machine [18]: there are no publicly available code from [18], making empirical comparison difficult. We discuss in the related work section the difference between their model and ours:
[18] uses a fixed size memory with an attention mechanism to move over the memory, while our model is based on a expandable memory controlled by a gating mechanism.

R2 and R5.
More details about optimization in the main paper: if the paper is accepted, we will move the optimization details from the supp. mat. to the main paper.

R2 and R4.
PTB experiments: more experiments are required to understand why we have a drop of performance compare to LSTM. Our current finding is that our model overfits (performances decrease when we have 500/100 hidden units with 100 stacks). Note that our first intention for this experiment was to check if our model behave reasonably on real data. We will add a discussion regarding this drop of performance.

R2. R2 has mostly positive comments.
- Memory as extra computation: Our work is still preliminary and we agree that in its current state, it is a fair criticism and we intend to address it in the future.
- Size of the stacks: In theory it is simple to make it dynamic, however for simplicity, we fixed that size in our code (to 500). We will add this information in the "implementation details" section.

R3. R3 has mostly positive comments.
- Concerning Mozer and Das (1993) and Zeng et al. (1994). We agree with R3 that our statement can be misleading and we will improve it in the final version of the paper.
- About discretization: We simply threshold the sigmoid to a 0-1 loss. We use it during validation to guide the network. We will describe this operation with more details if the paper is accepted.

R4. R4 has mostly positive comments.
R4 raises a very interesting point: There is a gap between toy and real data, and clever tricks such as (Le et al., 2015) or hyper-parameter search should be consider to help filling that gap. If the paper is accepted we will add a discussion of this issue.

R5.
Using a simpler model on the toy tasks: We focus on this slightly simpler version of our model to focus on our main contribution. We thought that this choice would make the results clearer to read, but we agree that it can be misinterpreted. Our model with the recurrent matrix still works on the tasks, and we will add experiments with the full model, if the paper is accepted.

R6:
- Comparison with Memory network [35]:
(1) There is no publicly available code for [35].
(2) Their model requires supervision over their controller (called "supporting facts" in [35]) , making the approach unusable in our unsupervised setting.
(3) In the supp. mat, there is a comparison with the Memory Network on their bAbI tasks (using their published numbers). We will move it to the main paper in the final version.
- Lack of acknowledgement for [18,35] compared to "previous work from the nineties": Our intention was to do a "reasonable job of pointing to related prior work" (R2) and to "[place this work] well in terms of past research" (R6), not to diminish the contributions of [18,35].
-"The authors pick a nice niche in this by focusing on stacks": (1) Das et al. (92) introduced the idea of using an external stack, not us. (2) Stacks are widely used and important data structure. (3) We consider that the simplicity of the stacks is an advantage as they are arguably easier to train.
- "Memory Networks [...] and the Neural Turing Machine [...] are strict generalizations of this paper": we strongly disagree with this statement. The Memory network requires supervision over its controller and does not build its memory (but use the input sequence as a fixed memory) while we both learn our controller and how to build the memory in a unsupervised way. For Neural Turing Machine, see above.
- "discuss the Memory Networks paper" : We mention it in the introduction and compare with it in the supp. mat. If the paper is accepted we will discuss the differences with our model in the related work section.
- This paper needs "more interesting experiments and motivation. The counting task seems very specifically tailored to this particular architecture": We also tackle tasks about copying, binary addition and language modeling (as well as multiplication and the bAbI tasks [35] in the supp. mat.). As noted by R2, "the exploration of the problem of binary addition and the Penn Treebank experiments in particular do indeed nicely and clearly differentiate the work here from prior art".